# MicroRNA-9 mediated the protective effect of ferulic acid on hypoxic-ischemic brain damage in neonatal rats

Keli Yao, Qin Yang, Yajuan Li, Ting Lan, Hong Yu*, Yang Yu*

Department of Histology and Embryology, School of Basic Medical Sciences, Southwest Medical University, Sichuan Province, China

* 759227112@qq.com (HY); yuyang80@swmu.edu.cn (YY)

## Abstract

Neonatal hypoxic-ischemic brain damage (HIBD) is prone to cognitive and memory impairments, and there is no effective clinical treatment until now. Ferulic acid (FA) is found within members of the genus *Angelica*, reportedly shows protective effects on neuronal damage. However, the protective effects of FA on HIBD remains unclear. In this study, using the Morris water maze task, we herein found that the impairment of spatial memory formation in adult rats exposed to HIBD was significantly reversed by FA treatment and the administration of LNA-miR-9. The expression of miRNA-9 was detected by RT-PCR analyses, and the results shown that miRNA-9 was significantly increased in the hippocampus of neonatal rats following HIBD and in the PC12 cells following hypoxic-ischemic injury, while FA and LNA-miR-9 both inhibited the expression of miRNA-9, suggesting that the therapeutic effect of FA was mainly attributed to the inhibition of miRNA-9 expression. Indeed, the silencing of miR-9 by LNA-miR-9 or FA similarly attenuated neuronal damage and cerebral atrophy in the rat hippocampus after HIBD, which was consistent with the restored expression levels of brain-derived neurotrophic factor (BDNF). Therefore, our findings indicate that FA treatment may protect against neuronal death through the inhibition of miRNA-9 induction in the rat hippocampus following hypoxic-ischemic damage.

## 1. Introduction

Neonatal hypoxic-ischemic brain damage (HIBD) is a severe disease that can cause irreversible neurological sequelae, such as cerebral palsy, mental deficiency, memory impairment and learning disabilities, and is often characterized by permanent neurological deficits [1, 2]. However, in some HIBD patients who receive early diagnosis and hypothermic treatments, still persist serious neurological sequelae. Currently, there is no effective clinical treatment for neurological sequelae induced by HIBD [3–5]. Thus, effective therapeutic agents which inhibit damage cascades activated after HIBD should be identified.

MicroRNAs are some small noncoding RNAs that regulate gene expression and suppression through several mechanisms. Recently microRNAs have been identified as critical

**Data Availability Statement:** All relevant data are within the manuscript and its Supporting Information files.

**Funding:** This work was supported by the Science and Technology Bureau of LuZhou City

(2017LZXNYD-J30).The funders had no role in study design, data collection and analysis, decision to publish, or preparation of the manuscript.

**Competing interests:** The authors have declared that no competing interests exist.

**Abbreviations:** HIBD, hypoxic-ischemic brain damage; HI, hypoxia-ischemia; FA, ferulic acid; SF, sodium ferulic; RT-PCR, real-time PCR; AchE, acetylcholinesterase; PSD-95, postsynaptic density protein-95; BDNF, brain-derived neurotrophic factor; miR-9, microRNA-9; LNA-miR-9, locked nucleic acid-miR-9; DG, dentate gyrus.

mediators of neuroinflammation and neurodegeneration in HIBD [6, 7]. Precious reports have demonstrated that miR-30b is upregulated during HIBD and is involved in the regulation of cellular apoptosis after injury [8]. MicroRNA-9 (miR-9) is highly expressed in the brain and has been implicated in the regulation of neurogenesis and proliferation as well as axonal development and neuronal migration [9–13]. As a result, miR-9 have unique advantages as targeted molecules in HIBD diagnosis and treatment. However, little attention been paid to the role of miR-9 in brain damage in infants caused by hypoxia.

Ferulic acid (FA) is a widely distributed constituent of plants which has been used for treating cerebrovascular diseases and cerebral ischemia by protecting neurons from degeneration and regulating inflammatory activity [14–16]. However, it is unknown whether the administration of FA has a neuroprotective effect on HIBD. Furthermore, the potential mechanisms remain unclear. Acetylcholinesterase (AchE), postsynaptic density protein-95 (PSD-95) involve in the progression of the transmission of nerve impulses,to evaluate the neuroprotective effect of FA, we currently examine the expression of AchE and PSD-95. In conclusion, we aimed to explore neuroprotective effect of FA on HIBD rats and further characterize the neuroprotective effect of FA on learning and memory ability, possibly via the downregulation of miR-9 following hypoxic-ischemic injury.

## 2. Methods and materials

### 2.1. Preparation of SF and LNA-miR-9

Sodium ferulic (SF, the sodium salt form of FA, purity >99.9%) was purchased from Dalian Meilunbio Biotechnology Co., Ltd. (CAS Registry Number: 24276-84-4, China), dissolved in 0.9% saline solution to a final concentration of 7 mg/ml, and subsequently stored at 4°C.

The sequence of LNA-miR-9 is 5'-tcatacaGCtAgAtaACcaAaGa-3', with the uppercase letters representing the sites of locked nucleic acid modifications [17]. LNA-miR-9 was synthesized by Sangon Biotech Co., Ltd. (Shanghai, China), dissolved in 0.01M PBS to a final concentration of 200 µM and stored at -20°C.

### 2.2. Cell culture and HI model establishment

PC12 cells (obtained from ATCC and provided by the Laboratory of Biochemistry and Molecular Biology, Southwest Medical University) were maintained in RPMI-1640 medium (Hyclone, USA) containing 10% equine serum (Solarbio, China), 5% fetal bovine serum (Solarbio, China) at 5% $CO_2$ and 37°C. Fresh medium was supplied every 48 h. The PC12 cells were divided into four groups: Control group, HI group, HI+SF group and HI+LNA group. In Control group, the PC12 cells were maintained in complete medium all the experiment time. To induce hypoxic-ischemic injury, cultured PC12 cells were serum-starved in RPMI-1640 medium supplemented with 1% equine serum and 1% fetal bovine serum and placed at 37°C in a humidified three-gas incubator (2% $O_2$, 93% $N_2$, and 5% $CO_2$; NUAER) for 2 h [18]. All experiments were repeated at least three times.

### 2.3. Animals and HIBD model establishment

One hundred and sixty-eight seven-day-old Sprague-Dawley (SD) rats, half male and half female, were provided by the Animal Department of Southwest Medical University (license number: SYXK (Sichuan) 2018–065, LuZhou City, China) and maintained under SPF conditions. The rats were divided into six groups randomly and treated in the following cohorts: In HIBD group, HIBD+SF group, HIBD+LNA group and HIBD+PBS group, the HIBD model was established by the classic method of Rice-Vannucci [19, 20]. In Sham group, the vagus

nerve was separated but not ligated, and the pups were placed in a similar container but exposed to normal room air. In Control group, the pups maintained under normal conditions routinely without surgery.

All animal experiments were performed under the protocol approved by the Animal Research Committee of Southwest medical university in accordance with the "Health Guide for the Care and Use of Laboratory Animals" which approved by Sichuan Experimental Animal Management Committee (Protocol Number: 202089). All surgery was performed under sodium pentobarbital anesthesia, and all efforts were made to minimize suffering.

## 2.4. Drug treatment

Considering that the robust neuroprotective effects of SF (0.5 μg/ml) and LNA-miR-9 (50μM/ml) were observed *in vitro* experiment, we decided to use those concentration in the following *in vitro* studies. After HI model established, in HI+SF group and HI+LNA group, the PC12 cells were treated immediately with SF or LNA-miR-9 for 24 h.

14 days after the HIBD model established, the rats in HIBD+SF group intraperitoneally injected with 50 mg/kg SF one time per day for 5 days and the drug concentration is calculated by the Animal equivalent dose calculation based on body surface area. The rats in HIBD+LNA group received a terminal intraperitoneal injection of sodium pentobarbital, 2 μl (400 μM) of LNA-miR-9 was stereotaxically microinjected into the lateral ventricle (1.0 mm posterior to bregma, -0.8 mm from the midline, and -3.5 mm deep from the dura) within 15 min. The rats in HIBD+PBS group received a terminal intraperitoneal injection of sodium pentobarbital, 2 μl of 0.01M PBS was stereotaxically microinjected into the lateral ventricle within 15min. LNA-miR-9 and PBS treatment was only administered for one day.

## 2.5. RNA, miRNA and protein isolation

Cell pellets were taken at 24 hours after drug treatment; the rats were killed at 26 days old and the hippocampus were dissected on ice, snap-frozen in liquid nitrogen immediately. RNA and miRNA were isolated from the same cell pellet and hippocampus tissue using miRcute-miRNA Extraction and Isolation kit (TIANGEN, china) following the manufacturer's protocol. Proteins were isolated from another cell pellet and hippocampus tissue by using RIPA buffer (high) (Solarbio). RNA quality was checked on a denaturing RNA Ethidium bromide (EB) gel (1% agarose, 110 V, 30 min). After proteins were isolated already, the protein content was determined by using BCA Protein Assay Kit (Solarbio).

## 2.6. Real-time PCR

cDNA synthesis from mRNA was done with a ReverTra Ace qPCR RT Master Mix Kit (FSQ-201, TOYOBO, Japan) according to the manufacturers' protocols (800ng total RNA per reaction). cDNA synthesis from miRNA was performed with a miRcute Plus miRNA First-Strand cDNA Kit (KR-211, TIANGEN) according to the manufacturer's instructions (1μg total RNA).

Real-time qPCR was performed with a SYBR® Green Realtime PCR Master Kit (TOYOBO) and miRcute Plus miRNA qPCR Kit (SYBR Green, TIANGEN) in a qTOWER³G device (Analytikjena, Germany). The specific PCR primers use for the detection of AchE, PSD-95 and β-actin were designed according to the NCBI sequence and synthesized by Sangon Biotech Co., Ltd. The sequences of the primers were as follows (Table 1):

**Table 1. Primer sequence.**

| Gene | Forward | Reverse |
|---|---|---|
| AchE | ACTGAACTACACCGTGGAGGAGAG | TTCAGGTTCAGGCTCACGTATTGC |
| PSD-95 | CAATGAAGTCAGAGCCCCCTA | CCTGCAACTCATATCCTGGGG |
| β-actin | CCCATCTATGAGGGTTACGC | TTTAATGTCACGCACGATTTC |
| miR-9 | TCTTTGGTTATCTAGCTGTATGA | Downstream primer provided by the kit. |
| U6 | GCTTCGGCAGCACATATACTAAAAT | Downstream primer provided by the kit. |

## 2.7. Western blot

Western blotting was performed as described previously [21]. 10μg of protein was loaded on 10% SDS-PAGE gel followed by transfer to PVDF membranes (Millipore, USA). Blots were blocked with 5% skim milk (5g milk powder dissolved in 100ml TBS-T) for 1 h at room temperature. After blocking, the membranes were incubated in specific antibody solutions (Rabbit monoclonal anti-AchE, 1:1000, Abcam-ab183591, USA; Rabbit monoclonal anti-PSD95, 1:3000, Abcam-ab76115, USA; Rabbit monoclonal anti-BDNF, 1:5000, Abcam-ab108319, USA; Rabbit polyclonal anti-β-actin, 1:10000, Proteintech-20536-1-AP, China) overnight at 4°C, followed by 1×TBS-T washing steps and then incubation with biotinylated antibody solutions (Proteintech, China) for 1 h. Blots were washed again and were placed into chemiluminescent HRP substrate (Millipore, USA) for 2 mins followed by observed by Clinxchemi Scope 6000 (Clinx, China). The absorbance values of the bands were quantified using an image analysis system and Image J. The protein levels were normalized against the β-actin intensity.

## 2.8. HE staining

Hematoxylin-eosin (HE) staining was performed on paraffin sections of brain tissue. Anesthetized rats were perfused with saline followed by 4% paraformaldehyde solution. The brains were then collected and immersed in 4% paraformaldehyde. After dehydration and vitrification, the brains were embedded in paraffin and cut into 6-μm sections from Bregma -2.16 mm to -4.20 mm. Serial sections (every 10th; intervals of 60-μm) were adhered to glass slides precoated with polyethylenimine. Paraffin-embedded brain sections were dewaxed with xylene and rehydrated with an ethanol series (absolute, 95%, 90%, 80%). The sections were then stained with HE, and images shown were acquired with Automatic digital slice scanning system (KFBio, China).

## 2.9. Morris water maze test

Four weeks after HIBD, 30-day-old rats were subjected to the Morris water maze test as described previously [22]. In brief, an open circular water-filled pool with a diameter of 160 cm, a height of 50 cm and a temperature of 22±1°C was prepared. The pool was divided into four quadrants, each of which had a reference object consisting of a different colored shape. In the experiment, animals were trained with no platform sessions to experience the pool environment (Day 0), and then hidden platform sessions were performed four times a day at 30-min intervals (Days 1–4). The platform was in the center of the NE quadrant and was placed 1.5~2 cm below the surface of the water during the hidden tests. Each trial had a limit of 60 sec, and the time to find the platform (escape latency) was recorded. At Day 5, after removing the platform, the probe test was conducted in which the time spent in each quadrant during 60 sec and the swimming route were recorded. The swimming behavior of the rats was recorded and tracked by an overhead video camera connected to a PC with an automated tracking software, and the data were analyzed using the behavior analysis system (Shanghai Xinruan Information Technology Co., Ltd., Shanghai, China).

## 2.10. Statistical analyses

All data are expressed as the means ± standard error (SE). Statistical analyses and statistical graph were performed using GraphPad Prism 7 software. Differences between two groups were evaluated statistically by using unpaired Student's t test. Differences among three or more groups were compared by analysis of variance (ANOVA) which with Tukey's adjustment for multiple comparisons. $P<0.05$ was considered statistically significant.

## 3. Result

### 3.1. FA ameliorated cerebral palsy, memory impairment and learning disabilities in HIBD rats

To determine whether FA can rescue HIBD-induced cerebral palsy, memory impairment and learning disabilities in neonatal rats, HE staining and the Morris water maze test were conducted.

Coronal section of the cerebral cortex and hippocampus showed that the area of the cerebral cortex, the size of the hippocampus and the layers of the dentate gyrus (DG) were dramatically

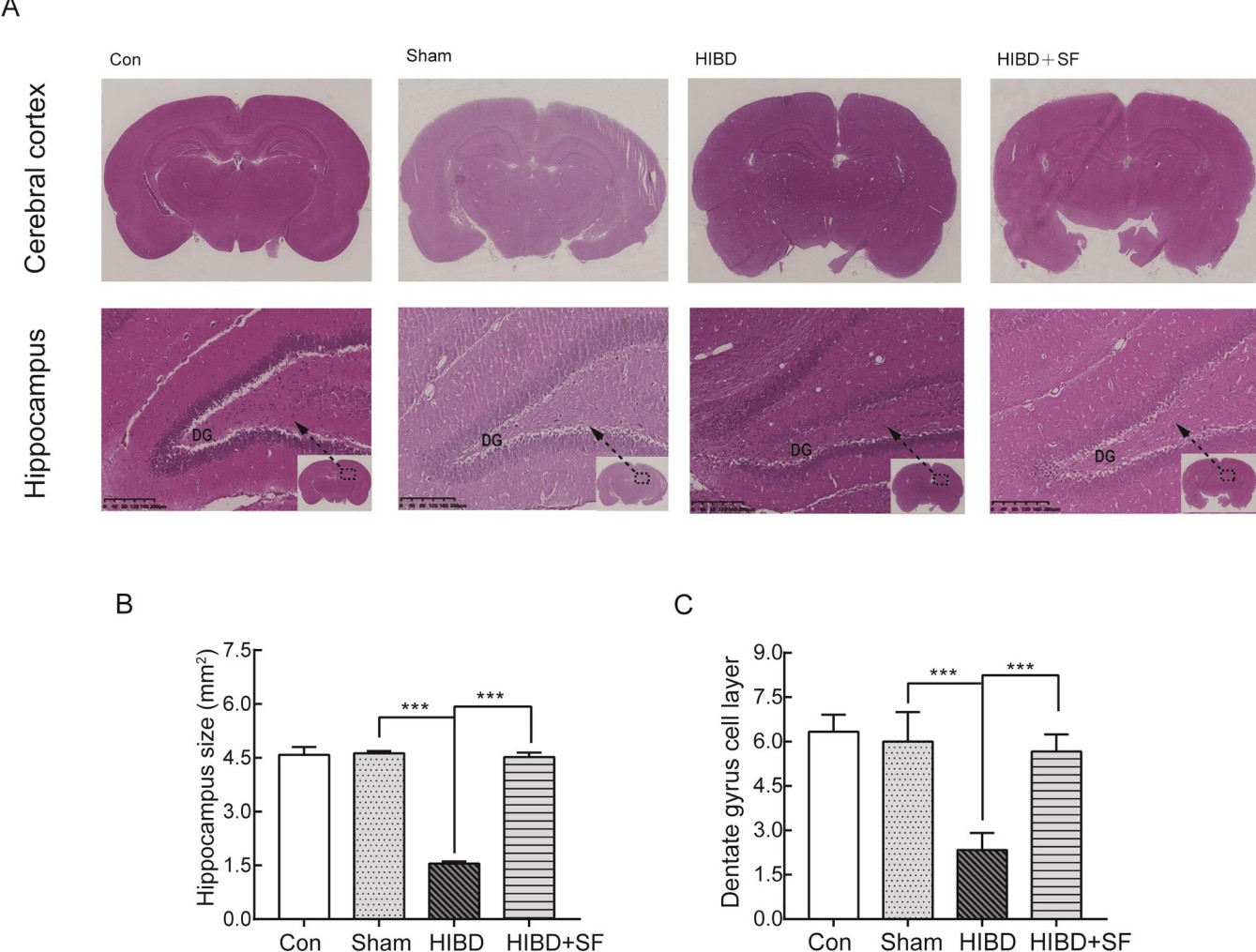

**Fig 1. FA ameliorated brain tissue loss in HIBD rats.** (A) Coronal sections stained with hematoxylin-eosin are shown. (B, C) Quantification of the size of the hippocampus and the layers of the hippocampal dentate gyrus in the different groups. **** $P < 0.0001$, *** $P < 0.001$. N = 6 for HE staining.

decreased in the HIBD group compared with the sham group, and the administration of FA significantly suppressed the decrease in hippocampal neurons, the area of cerebral cortex, the size of the hippocampus and the layers of the DG (Fig 1A–1C, $P < 0.001$ vs. HIBD).

To evaluate long-term spatial learning and memory ability, the Morris water maze was performed on 35-day-old rats after SF application. In the acquisition trial, HIBD rats showed decreased spatial learning ability with a significantly increased escape latency compared with those of the control and sham groups on the training days. Importantly, as shown in Fig 2A, daily SF treatment significantly ameliorated the impairment induced by HIBD ($P < 0.001$). In the retention trial, a significant reduction in the exploration distance in the target quadrant was observed in HIBD rats compared with HIBD+SF rats (Fig 2B and 2C, $P < 0.001$). These results further confirmed that spatial memory retrieval was impaired after HIBD and that FA treatment succeeded in preventing this impairment.

Taken together, these results indicate that FA treatment alleviates deficits in ischemia-induced cell death and learning and memory function in neonatal rats after HIBD.

## 3.2. The expression of miR-9 significantly increased under hypoxic-ischemic conditions, and FA attenuated the expression of miR-9

To establish whether miR-9 is involved in HIBD, we investigated the expression levels of miR-9 in the PC12 cells and hippocampus of HI condition compared with control using RT-PCR analysis. The RT-PCR results showed that the expression level of miR-9 in PC12 cells was markedly higher after HI injury than in the control groups (Fig 3A, $P < 0.05$). Similarly, miR-9 expression was almost doubled in the hippocampus of hypoxic-ischemic rats compared to

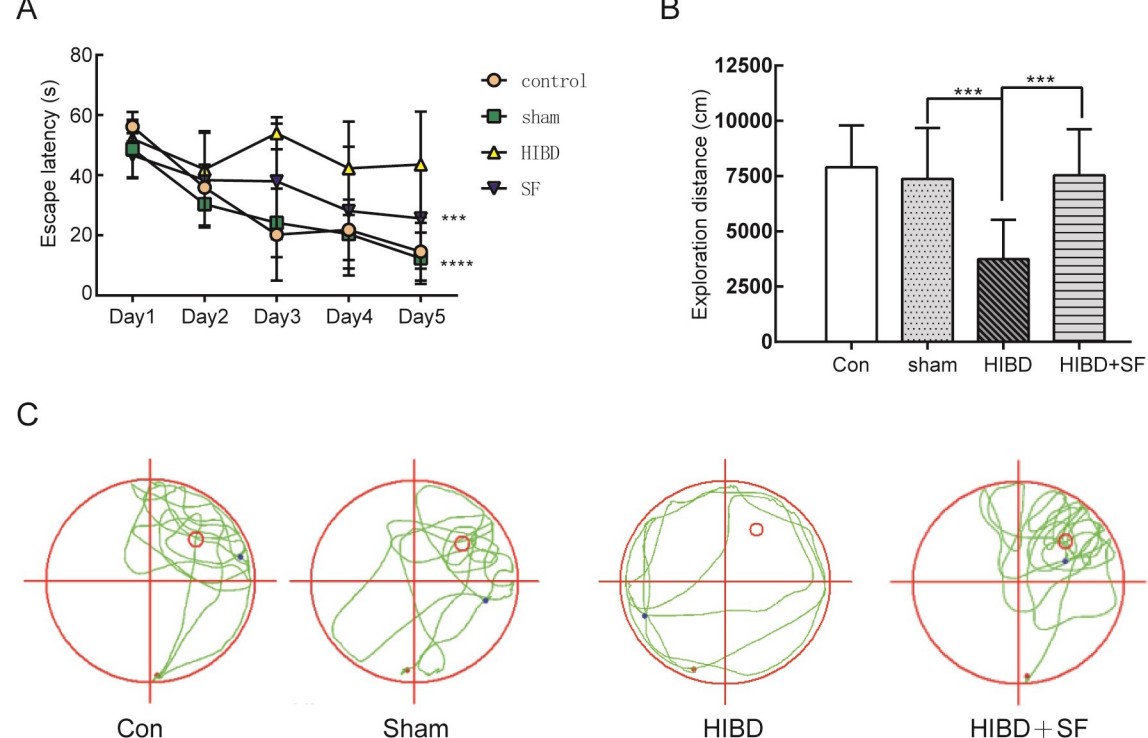

**Fig 2. FA ameliorated learning and memory impairments in HIBD rats.** (A) The escape latency (T value) of the rats in the training period. (B) The exploration distance (P value) of the rats in the exploratory trials. (C) The exploration route in the retention trial. **** $P < 0.0001$, *** $P < 0.001$. N = 10 for Morris water maze analysis.

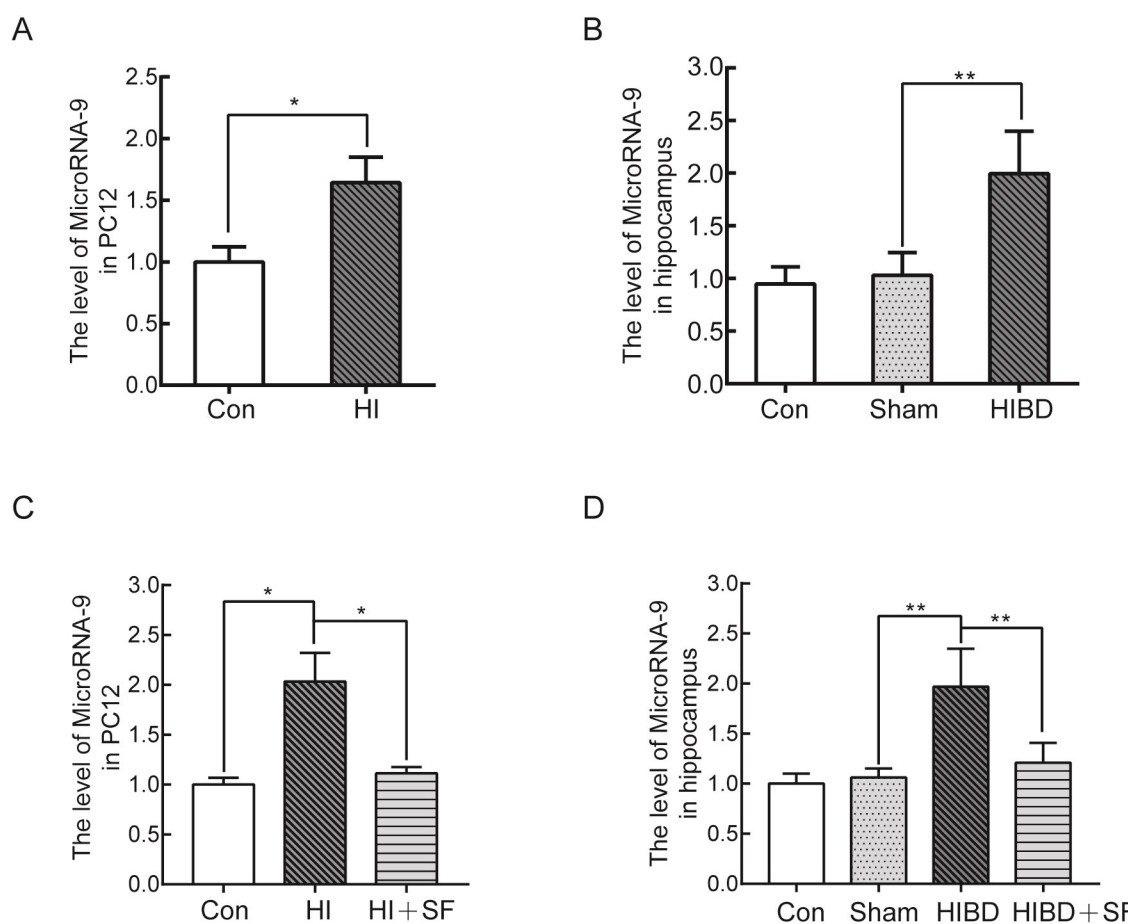

**Fig 3. FA attenuated the upregulation of miR-9 under hypoxic-ischemic conditions in cultured PC12 cells and the rat hippocampus.** (A) RT-PCR analysis of miR-9 expression in PC12 cells after HI. (B) RT-PCR analysis of miR-9 expression in the rat hippocampus after HIBD. ** $P < 0.01$, * $P < 0.05$ vs. control. (C) RT-PCR analysis of miR-9 expression in PC12 cells after FA treatment. (D) RT-PCR analysis of miR-9 expression in the rat hippocampus after FA treatment. ** $P < 0.01$, * $P < 0.05$ vs. HIBD. N = 4 for cell analysis and N = 6 for animal analysis.

sham rats ([Fig 3B], $P < 0.01$). To further determine the therapeutic effects of FA on HIBD, we examined the expression levels of miR-9 after treating the rats with SF daily for 5 days and then compared them with the levels in the sham group. In PC12 cells cultured under serum-oxygen deprivation conditions for 2 h, miR-9 release was significantly reduced after treatment with SF ([Fig 3C], $P < 0.05$). Similar results were also observed in HIBD rats, and the expression levels of miR-9 were markedly lower in the HIBD+SF group than in the HIBD group ([Fig 3D], $P < 0.01$). These results further confirmed that miR-9 plays an important role in neonatal HIBD and that the neuroprotective effect of FA may occur through the downregulation of the expression levels of miR-9.

### 3.3. Inhibition of miR-9 protected against hypoxic-ischemic brain damage in neonatal rats

To further reveal the role of miR-9 following hypoxic-ischemic damage. We investigated whether inhibiting the expression of miR-9 in vivo protects hippocampal neurons. We injected LNA-miR-9 (400 μM) into the lateral ventricle of 21-day-old neonatal rats after HIBD. LNA-miR-9 treatment significantly reduced brain injury in the rats, which resulted in a significant

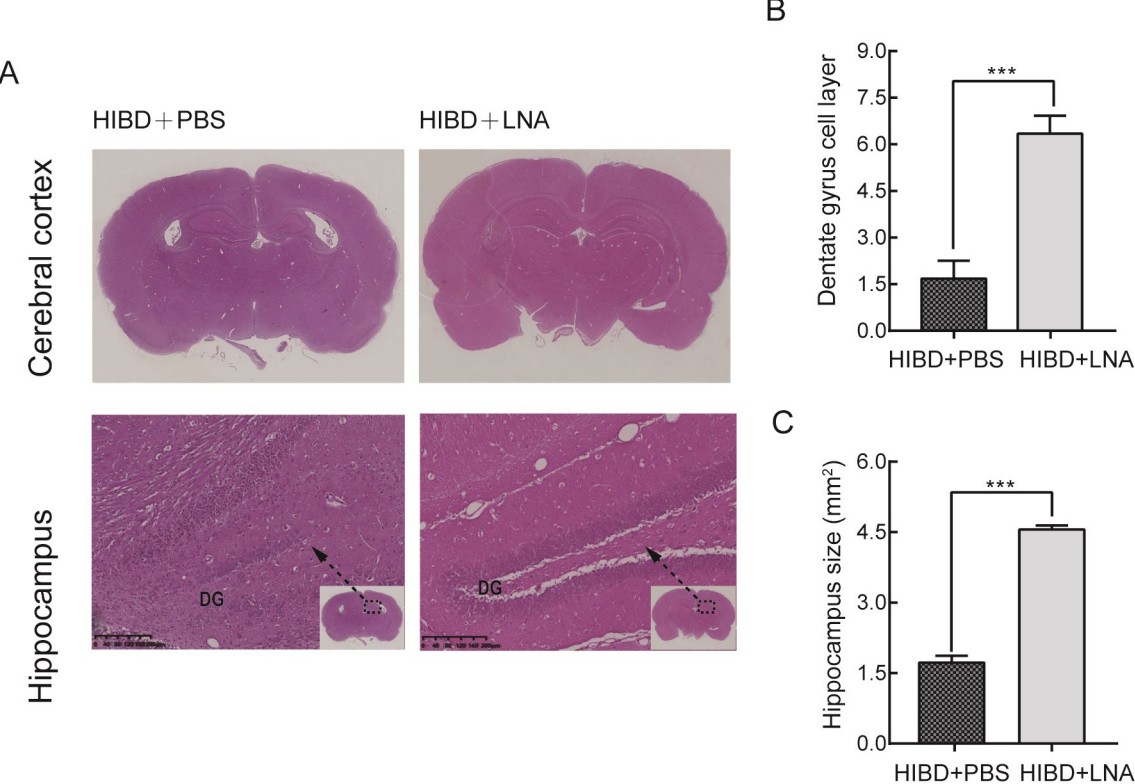

**Fig 4. LNA-miR-9 attenuated brain tissue loss in HIBD rats.** (A) Coronal sections stained with hematoxylin/eosin are shown. (B, C) Quantification of the size of the hippocampus and the layers of hippocampal dentate gyrus in the different groups. *** $P < 0.001$ vs HIBD+PBS. N = 6 for HE staining.

increase in the surviving brain volume and cell layers of the DG compared to that of the HIBD +PBS group (Fig 4A–4C, $P < 0.001$). The Morris water maze was performed on 35-day-old rats after LNA-miR-9 application.

In the acquisition trial, HIBD+PBS rats showed decreased spatial learning ability with a significantly increased escape latency. Notably, LNA-miR-9 treatment significantly ameliorated the impairments of HIBD+PBS rats (Fig 5A, $P < 0.001$). In the retention trial, a significant reduction in the exploration distance in the target quadrant was observed in HIBD+PBS rats compared with HIBD+LNA rats (Fig 5B and 5C, $P < 0.001$). These results indicated that LNA-miR-9 protected against brain injury induced by HIBD in rats.

## 3.4. FA and LNA-miR-9 protected against synaptic marker loss induced by HI

AchE is a polymorphic enzyme commonly known for its cholinergic action that stops cholinergic signaling in the brain by hydrolyzing acetylcholine to choline and acetate [23]. The expression of AchE is closely correlated with the neuroinflammatory response and neurological dysfunction [24, 25]. Studies have shown that the expression of AchE is increased after hypoxia-ischemia (HI) in newborn rats. Further research has confirmed that the inhibition of the AchE receptor reduces brain damage after HI [26]. Previous reports have indicated that postsynaptic density protein (PSD)-95 expression is markedly decreased in the hippocampal CA1 region after brain injury [27–29]. Therefore, we currently investigated neuroprotective effect of FA on HIBD in rats by examining the expression of AchE and PSD-95.

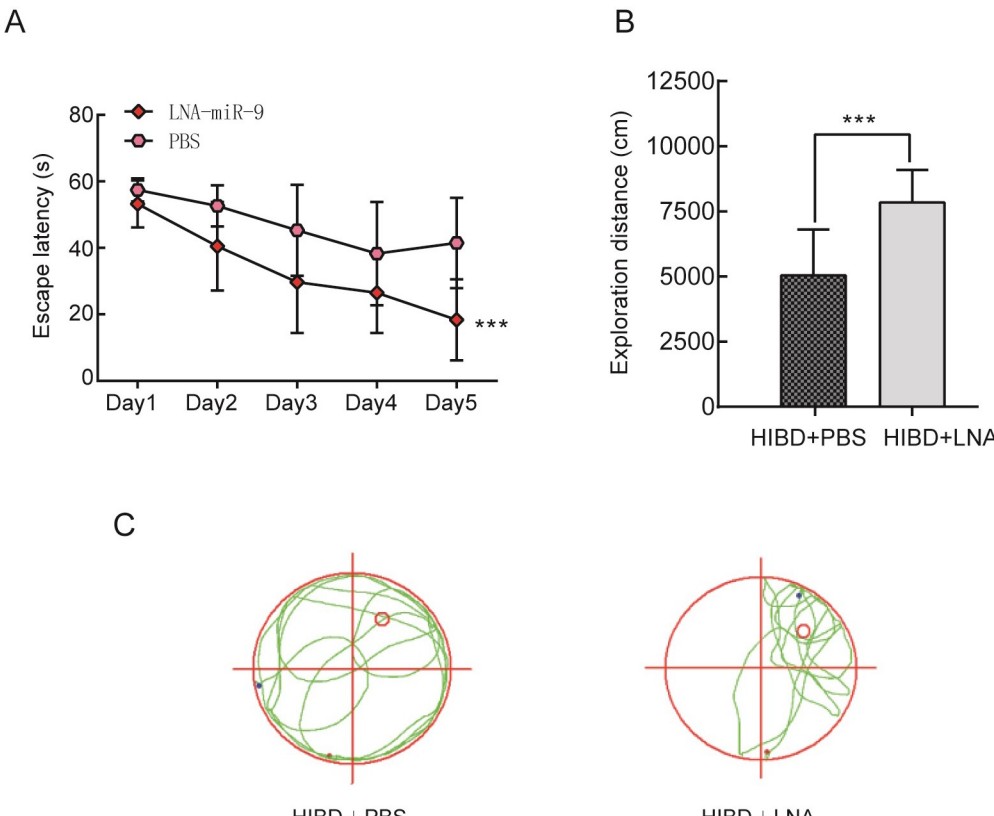

**Fig 5. LNA-miR-9 attenuated learning and memory impairments in HIBD rats.** (A) The escape latency (T value) of the rats in the training period. (B) The exploration distance (P value) of the rats in the exploratory trials. (C) The exploration route in the retention trial. *** $P < 0.001$ vs HIBD+PBS. N = 10 for Morris water maze analysis.

As shown in Fig 6A and 6B, the expression of AchE mRNA in the PC12 cells and rat hippocampus was significantly higher in the HI group and HIBD group ($P < 0.01$ vs HI and $P < 0.05$ vs HIBD). SF and LNA-miR-9 notably decreased the HI-induced upregulation of AchE mRNA ($P < 0.01$ vs HI, $P < 0.05$ vs HIBD and $P < 0.01$ vs HIBD+PBS). On the other hand, hypoxia-ischemia significantly reduced the expression of PSD-95 mRNA in the PC12 cells and rat hippocampus ($P < 0.05$ vs Con and Sham). After SF and LNA-miR-9 treatment, the expression of PSD-95 mRNA increased (Fig 6C and 6D, $P < 0.01$ vs HI, $P < 0.05$ vs HIBD and HIBD+PBS).

In addition, Western blot analysis was performed to verify the protein expression levels of AchE and PSD-95 in the PC12 cells and rat hippocampus. The results shown in Fig 7A, 7B, 7C and 7D suggest that hypoxia-ischemia significantly increased the protein levels of AchE and decreased the levels of PSD-95 compared with those in the control group and sham group ($P < 0.01$). Meanwhile, treatment with SF and LNA-miR-9 reduced the protein levels of AchE and upregulated the protein expression of PSD-95 ($P < 0.01$ vs HI, $P < 0.01$ vs HIBD and HIBD+PBS). Collectively, these results indicated that miR-9 mediated the neuroprotective effect of FA against HIBD.

## 3.5. BDNF signaling participated in the neuroprotective effects of FA in HIBD neonatal rats

Brain-derived neurotrophic factor (BDNF) has been recognized as a protective effector of synaptic transmission and cognitive function after HI brain injury [30]. Previous reports have shown that

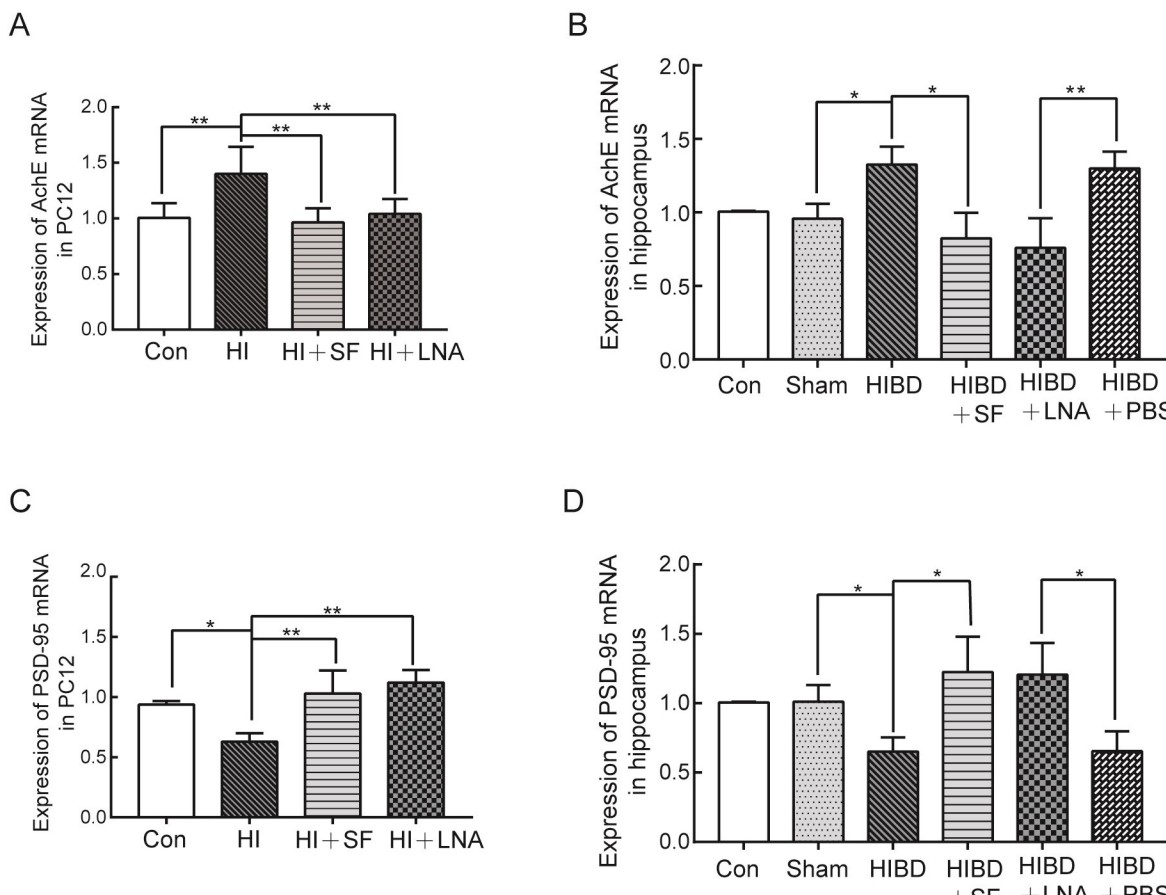

**Fig 6. RT-PCR analysis of the gene expression of AchE mRNA and PSD-95 mRNA in PC12 cells and rat hippocampus.** SF and LNA effectively reversed the expression changes in AchE mRNA and PSD-95 mRNA caused by hypoxic-ischemic injury. (A) The expression of AchE mRNA in PC12 cells. (B) The expression of AchE mRNA in the rat hippocampus. (C) The expression of PSD-95 mRNA in PC12 cells. (D) The expression of PSD-95 mRNA in the rat hippocampus. ** $P < 0.01$, * $P < 0.05$. N = 4 for cell analysis and N = 6 for animal analysis.

the expression levels of BDNF are directly regulated by miRs and that miR-9 plays an important role in the regulation of axonal projections in response to BDNF signaling [31]. Therefore, we examined whether BDNF signaling functions in mediating miR-9 expression in HIBD neonatal rats. Western blot analysis revealed that BDNF expression was significantly decreased in the hippocampus of the rats in the HIBD group compared with those in the sham group after hypoxic-ischemic damage ($P <0.01$). In the neonatal rats in the HIBD+SF and HIBD+LNA-miR-9 groups, increased BDNF protein expression was observed (Fig 8A, $P <0.01$ vs HIBD and HIBD+PBS). These results confirmed that FA regulated miR-9 expression by increasing the release of BDNF.

## Discussion

HIBD is one of the major causes of neonatal death and disability in human neonates worldwide [32]. It is urgent to identify drugs that protect against neuronal damage to reduce the neurological disorder induced by HIBD. In this study, we demonstrated for the first time that FA can improve learning and memory impairment caused by HIBD in rats. The neuroprotective effects of FA are likely the result of the regulation of miR-9 expression through increased release of BDNF. We established HIBD model used on P7 rats, because P7 rats correspond to human neonatal [33]. However, it is difficult to inject LNA-miR-9 with a stereotaxic

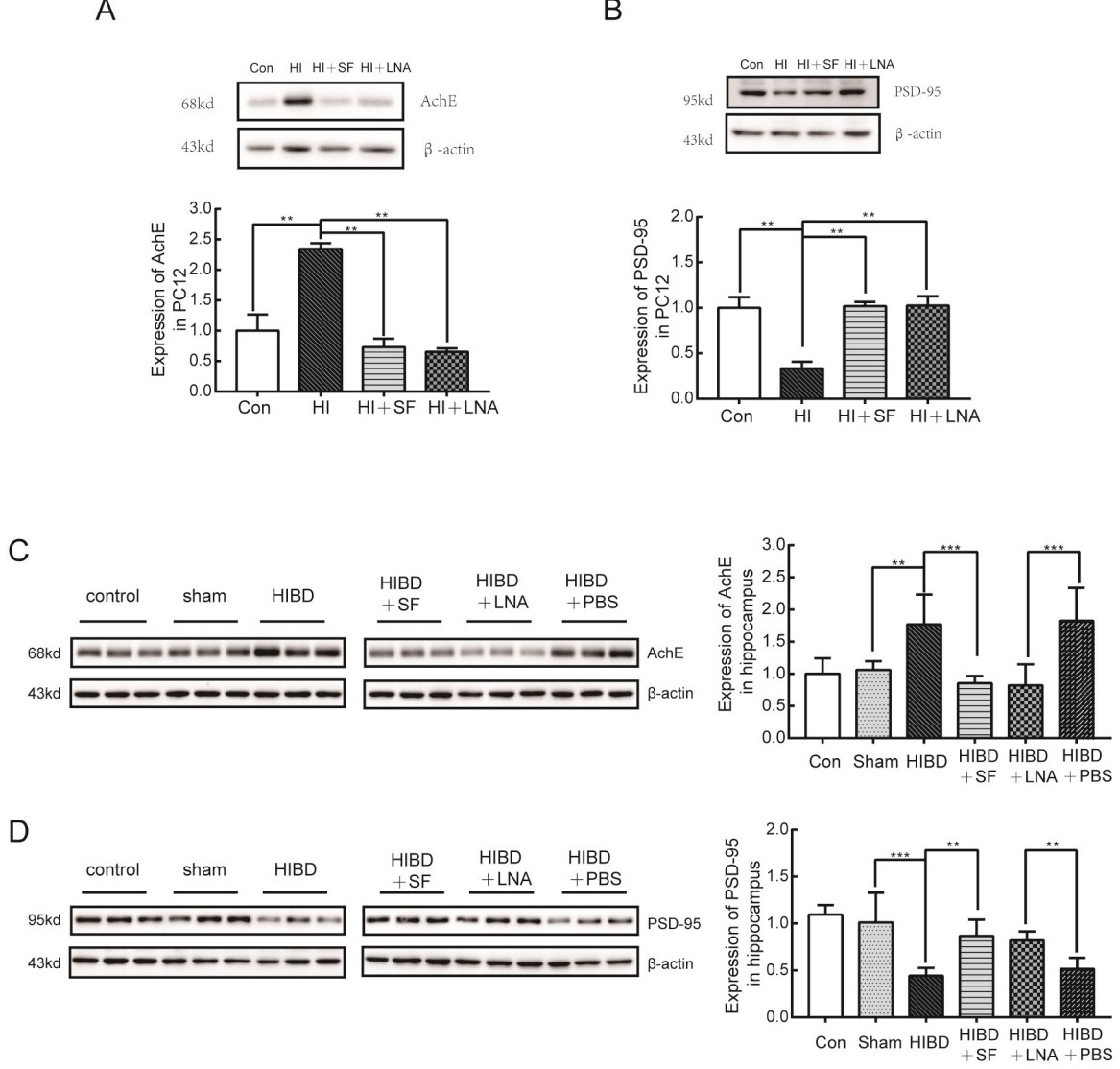

**Fig 7. Western blot analysis of the protein expression of AchE and PSD-95 in PC12 cells and rat hippocampus.** SF and LNA-miR-9 effectively reversed the expression changes in AchE and PSD-95 caused by ischemic-hypoxic injury. (A) The expression of AchE in PC12 cells. (B) The expression of AchE in the rat hippocampus. (C) The expression of PSD-95 in PC12 cells. (D) The expression of PSD-95 in the rat hippocampus. $^{***}$ $P < 0.001$, $^{**}$ $P < 0.01$ vs HIBD or HIBD+PBS. N = 4 for cell analysis and N = 6 for animal analysis.

instrument on the P7 damaged brain. P21 rats are in the stage of high-speed brain growth and development, and drug intervention at this stage can observe the effect of FA and LNA-miR-9 on the neurological sequelae effectively.

A growing number of researchers have discovered the beneficial influence of FA in stress-induced injury [34]. Treatment with FA has a protective effect by reducing oxidative stress and inhibiting ROS production in PC12 cells treated with the neurotoxin lead acetate [35]. Consistently, a previous report also revealed that FA possesses the ability to inhibit LPS-induced neuroinflammation [36]. However, little is known about its protective effects in HI brain injury. Previous reports have indicated that hypoxia-ischemia can reduce the long-term spatial learning and memory ability of rats [37]. Our results showed that the size and neurons number of cerebral cortex and hippocampus were increased compared with those of the HIBD group and the

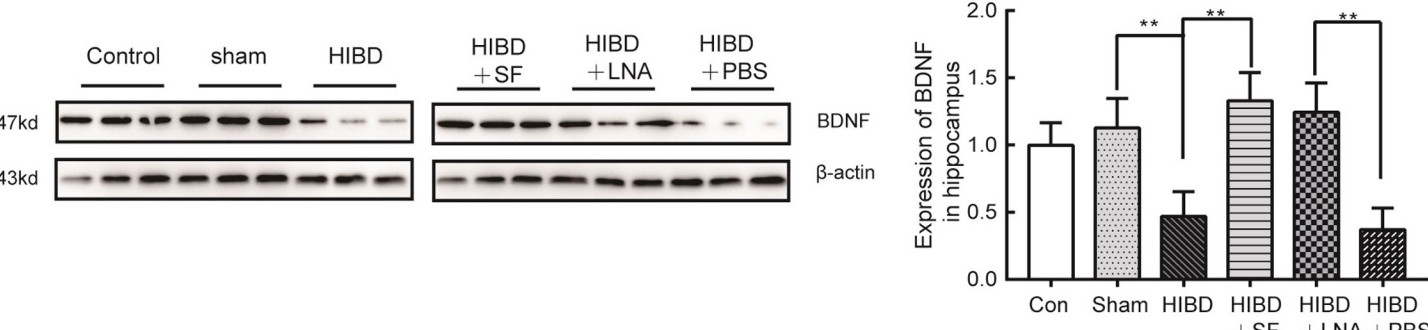

**Fig 8. Western blot analysis of the protein expression of BDNF in the rat hippocampus.** SF and LNA-miR-9 effectively increased the expression of BDNF in HIBD. (A) The expression of BDNF in rat hippocampus. ** $P < 0.01$ vs HIBD or HIBD+PBS. N = 6 for animal analysis.

HIBD+PBS group after treatment with SF and LNA-miR-9 by HE staining, indicated that FA can relieve the damage of brain significantly. Behavioral experiments showed that after treatment with SF and LNA-miR-9, the T value was significantly shortened and the P value was significantly prolonged compared with those of the HIBD group and HIBD+PBS group, suggesting that FA can improve the spatial learning and memory ability of HIBD rats.

MicroRNAs are small, noncoding RNAs that negatively regulate the expression of target mRNAs through degradation and translational inhibition [38]. An increasing number of studies have shown that microRNAs play an important regulatory role in neurodevelopment, metabolism and cancer [39–41]. miR-9 is a highly expressed miRNA in the brain and is enriched in synapses. Studies have shown that miR-9 can promote proliferation, migration, differentiation and apoptosis and regulate the dendrites of neurons [42–45]. However, whether miR-9 is involved in hypoxic-ischemic injury and recovery in the rat brain has not been clearly reported. This study found that the expression of miR-9 in PC12 cells and the hippocampus of the rats in the model group was significantly upregulated compared with that in the normal group, indicating that ischemia-hypoxia can influence the level of miR-9 expression in the process of neuronal damage. In addition, after SF treatment, the expression of miR-9 was significantly lower, revealing that FA improve the learning and memory ability of HIBD rats by downregulating the expression of miR-9.

BDNF is broadly expressed in the developing mammalian brain, and evidence has shown that hypoxia-ischemia in rats results in increased BDNF expression in neurons [46]. Our experiments found that BDNF expression was significantly increased after treatment with FA and the inhibition of the expression levels of miR-9 in HIBD. In addition, the inhibition of miR-9 can downregulate the expression of AchE and upregulate the expression of PSD-95. The mechanism may be that the increased release of BDNF can protect against neuronal death in the rat hippocampus. This ensures the normal transmission of neuronal signals, thus effectively inhibiting nerve damage after hypoxia. In addition, there may be other mechanisms by which FA improves the learning and memory ability of rats with hypoxic-ischemic injury. Further research is needed to clarify the mechanism underlying the effect of FA in the treatment of HIBD and make better use of it in the clinic; traditional Chinese medicine containing FA provides a theoretical basis for the treatment of HIBD.

## Supporting information

**S1 Checklist The ARRIVE guidelines checklist.**
(DOCX)

**S1 Fig.**
(TIF)

**S1 Original images.**
(PDF)

## Acknowledgments

We thank the Southwest Medical University Laboratory of Biochemistry and Molecular Biology for technical assistance and the Public Health Center for assistance and use of the Morris water maze.

## Author Contributions

**Conceptualization:** Hong Yu, Yang Yu.

**Data curation:** Keli Yao.

**Formal analysis:** Keli Yao.

**Investigation:** Keli Yao, Qin Yang, Ting Lan.

**Methodology:** Keli Yao, Yajuan Li.

**Project administration:** Keli Yao.

**Validation:** Keli Yao.

**Writing – original draft:** Keli Yao.

**Writing – review & editing:** Hong Yu, Yang Yu.

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
