## [Decision Letter · Decision Letter 0]

12 Mar 2020

PONE-D-20-01760

MicroRNA-9 Mediated the Protective Effect of Ferulic Acid on Hypoxic-Ischemic Brain Injury in Neonatal Rats

PLOS ONE

Dear Ms Yao,

Thank you for submitting your manuscript to PLOS ONE. After careful consideration, we feel that it has merit but does not fully meet PLOS ONE’s publication criteria as it currently stands. Therefore, we invite you to submit a revised version of the manuscript that addresses the points raised during the review process.

We would appreciate receiving your revised manuscript by Apr 26 2020 11:59PM. To enhance the reproducibility of your results, we recommend that if applicable you deposit your laboratory protocols in protocols.io, where a protocol can be assigned its own identifier (DOI) such that it can be cited independently in the future. For instructions see: http://journals.plos.org/plosone/s/submission-guidelines#loc-laboratory-protocols

We look forward to receiving your revised manuscript.

Kind regards,

Giuseppe Pignataro, MD, PhD

Academic Editor

PLOS ONE

Journal Requirements:

2. As part of your revision, please complete and submit a copy of the ARRIVE Guidelines checklist, a document that aims to improve experimental reporting and reproducibility of animal studies for purposes of post-publication data analysis and reproducibility: https://www.nc3rs.org.uk/arrive-guidelines. Please include your completed checklist as a Supporting Information file. Note that if your paper is accepted for publication, this checklist will be published as part of your article. As part of your revisions please be sure to specify the number of animals used in your study, including details of any animals that were excluded from your study for any reason (including death).

3. Please include in Methods section the source of the cell lines.

5. We note you have included a table to which you do not refer in the text of your manuscript. Please ensure that you refer to Table 1 in your text; if accepted, production will need this reference to link the reader to the Table.

Reviewers' comments:

Reviewer's Responses to Questions

**Comments to the Author**

1. Is the manuscript technically sound, and do the data support the conclusions?

Reviewer #1: Yes

2. Has the statistical analysis been performed appropriately and rigorously? 

Reviewer #1: Yes

3. Have the authors made all data underlying the findings in their manuscript fully available?

Reviewer #1: Yes

4. Is the manuscript presented in an intelligible fashion and written in standard English?

Reviewer #1: Yes

5. Review Comments to the Author

Reviewer #1: The paper authored by Keli Yao, and colleagues investigates the neuroprotective effects of Ferulic acid (FA) on learning and memory improvement in rats subjected to neonatal hypoxic-ischemic brain damage (NHIBD). They hypothesized that neuroprotective effects of FA are likely the result of the regulation of miR-9 expression through increased release of BDNF. In particular, researchers demonstrated that the intraperitoneally administration of Sodium ferulic (50 mg/kg) into the rats starting from to postnatal day twenty-one, one time per day for 5 days, induced the reduction hippocampal damage and prevented impairment in spatial learning and memory ability, evaluated by using Morris water maze test. In addition, they showed that the expression levels of miR-9 was significantly increased in the hippocampus of hypoxic-ischemic rats compared to sham rats. Moreover, they demonstrated that there was a significant reduction of the expression levels of miR-9 after treating the rats with SF daily for 5 days and then compared them with the levels in the sham group. In the same way, in vitro model, they observed in PC12 cells cultured under serum-oxygen deprivation conditions for 2 hours, that miR-9 release was significantly reduced after treatment with SF. Furthermore, they showed that the inhibition of miR-9, by using of LNA-miR-9, protected the hippocampus and ameliorated the spatial learning ability of rats subjected to NHIBD. They also observed, in the rat hippocampus and PC12 cells, that the treatment with SF and LNA-miR-9 determined the reduction of expression levels of AchE mRNA and protein, and induced the increase of expression levels of PSD-95 mRNA and protein. In the last part of work, the authors demonstrated that expression levels of BDNF protein significantly decreased in the hippocampus of the rats in HIBD group compared with those in the sham group after hypoxic-ischemic damage, conversely, in HIBD+SF and HIBD+LNA-miR-9 groups, increased BDNF protein expression had been observed.

Although the manuscript technically sounds, experiments have been performed with rigor, through appropriate controls, replication and sample size and the data produced support the conclusions, some points should be improved to reinforce the significance of the paper.

1. In the “discussion” section, it is necessary to explain, why they choose the dose of Sodium ferulic (50 mg/kg) for the intraperitoneally administration into the rats, since the mentioned papers used different dosage protocols.

2. In the “discussion” section, it is necessary to explain why they choose to start FA 21 days postnatally. It would be appropriate to understand whether the administration of FA starting from the phases immediately subsequent the induction of the damage, may also determine a greater neuroprotective effect in animals.

3. Authors should show the effect of FA administration in the cortex, since cortex is one of the main damaged areas after an hypoxic-ischemic damage.

4. In ”methods” section, it is appropriate to indicate the total number of rats were used in the paper.

5. In ”methods” section, it is appropriate to indicate the total number of slices for each mouse used to perform the H-E staining, also reporting the bregma coordinate.

6. PLOS authors have the option to publish the peer review history of their article (what does this mean?). If published, this will include your full peer review and any attached files.

Reviewer #1: No

---

## [Author Response · Author response to Decision Letter 0]

16 Apr 2020

Reviewer #1:

1. Response to comment: Why we choose the dose of Sodium ferulic (50 mg/kg) for the intraperitoneally administration into the rats.

Response: The dose of Sodium ferulic (50 mg/kg) is calculated by the Animal equivalent dose calculation based on body surface area. The calculation procedure is: 

AED (mg / kg) = Human does (mg / kg) × Km ratio = 8.1 (mg / kg) × (37/6) = 50 mg / kg in rats. We have supplemented these data in the “methods and materials” section.

2. Response to comment: Why we choose to start FA at P21 rats.

Response: Since it is difficult to inject LNA-miR-9 with a stereotaxic instrument on the P7 damaged brain. 21-day-old rats are in the stage of high-speed brain growth and development, and drug intervention at this stage can observe the effect of FA and LNA-miR-9 on the neurological sequelae effectively. In addition, intraperitoneal injection of SF and brain stereotactic injection of LNA-miR-9 need to be synchronized at the same time, and the surgery of brain stereotactic injection is an invasive craniotomy, considering that rats are weaned at 21 days postnatally and have ability to eat independently, thus we choose to start drug treatment at P21 rats.

3. Response to comment: About “Authors should show the effect of FA administration in the cortex”.

Response: HE staining showed changes in the rat cerebral cortex and hippocampus, but the changes in the cerebral cortex were far less than those of in the rat hippocampus. Furthermore, since we mainly studied learning and memory impairment caused by HIBD, which is closely related to the hippocampus, the results of the cerebral cortex changes are only shown by HE staining. Relevant content about cerebral cortex have been supplemented in the paper.

4. Response to comment: About “indicate the total number of rats were used in the paper”.

Response: In this experiment, we used 168 rats, which have been supplemented in the “methods and materials” section.

5. Response to comment: About “indicate the total number of slices for each mouse used to perform the H-E staining, also reporting the bregma coordinate”.

Response: In this experiment, the brains were cut into 6-μm sections from Bregma -2.16 mm to -4.20 mm. Series of each 10 consecutive sections (intervals of 60μm) were cut and stained with HE.

Other changes: In the “Introduction” section and the “methods and materials” section, several statements that we made were more ambiguous than intended, and we have adjusted to the text to be clearer. We tried our best to improve the manuscript and made some changes in the manuscript. These changes will not influence the content and framework of the paper. And here we did not list the changes but marked in red in revised paper.

---

## [Decision Letter · Decision Letter 1]

14 May 2020

MicroRNA-9 Mediated the Protective Effect of Ferulic Acid on Hypoxic-Ischemic Brain Damage in Neonatal Rats

PONE-D-20-01760R1

Dear Dr. Yao,

We are pleased to inform you that your manuscript has been judged scientifically suitable for publication and will be formally accepted for publication once it complies with all outstanding technical requirements.

With kind regards,

Giuseppe Pignataro, MD, PhD

Academic Editor

PLOS ONE

Additional Editor Comments (optional):

Reviewers' comments:

Reviewer's Responses to Questions

**Comments to the Author**

1. If the authors have adequately addressed your comments raised in a previous round of review and you feel that this manuscript is now acceptable for publication, you may indicate that here to bypass the “Comments to the Author” section, enter your conflict of interest statement in the “Confidential to Editor” section, and submit your "Accept" recommendation.

Reviewer #1: (No Response)

2. Is the manuscript technically sound, and do the data support the conclusions?

Reviewer #1: Yes

3. Has the statistical analysis been performed appropriately and rigorously? 

Reviewer #1: Yes

4. Have the authors made all data underlying the findings in their manuscript fully available?

Reviewer #1: Yes

5. Is the manuscript presented in an intelligible fashion and written in standard English?

Reviewer #1: Yes

6. Review Comments to the Author

Reviewer #1: The manuscript technically sounds, experiments have been performed with rigor, through appropriate controls, replication and sample size and the data produced support the conclusions.

The authors answered to comments and made the necessary changes to improve the manuscript.

I accepted the revision of the paper by Keli Yao and colleagues.

However, I would like to precise that, although the paper has been generally improved and is now suitable for publication, it is necessary to introduce these modifies before accepting it for publication:

1) In the revised paper, in the “introduction” section, in the sentence “In conclusion, we aimed to explore neuroprotective effect of FA on HIBD patients and …” (this sentence have been highlighted by authors by using red text to indicate the changes made). It is necessary that the authors correct the word “patients” with rats, because the study has been performed in rats and not in patients.

2) In the revised paper, in the “methods” section, in the sentence “The rats were divided into six groups randomly and treated in the following cohorts: In HIBD group, HIBD+SF group, HIBD+LNA group and HIBD+PBS group, the HIBD model was established by the classic method of Rice-Vannucci (this sentence have been highlighted by authors by using red text to indicate the changes made). It is necessary that the authors, indicate that it has been performed a few modification, because they reported in the next sentence that: ”The left carotid artery of 7-day-old SD rats was permanently unilateral

ligated and removed,…”(this sentence was already present in the first version of paper).

3) In the revised paper, in the “methods” section, It is necessary that the authors, indicate the gender of rats used in the procedure.

4) In the revised paper, in the “discussion” section, in the sentence “We established HIBD model used on P7 rats, because P7 rats correspond to human neonatal” (this sentence have been highlighted by authors by using red text to indicate the changes made). It is necessary that the authors improve the second part of this affirmation and add a reference.

7. PLOS authors have the option to publish the peer review history of their article (what does this mean?). If published, this will include your full peer review and any attached files.

Reviewer #1: No

---

## [Editor Report · Acceptance letter]

20 May 2020

PONE-D-20-01760R1 

MicroRNA-9 Mediated the Protective Effect of Ferulic Acid on Hypoxic-Ischemic Brain Damage in Neonatal Rats 

Dear Dr. Yao:

I am pleased to inform you that your manuscript has been deemed suitable for publication in PLOS ONE. Congratulations! Your manuscript is now with our production department. 

With kind regards,

on behalf of

Prof. Giuseppe Pignataro 

Academic Editor

PLOS ONE